# Disparities in Breast Cancer Care—How Factors Related to Prevention, Diagnosis, and Treatment Drive Inequity

**DOI:** 10.3390/healthcare12040462

**Published:** 2024-02-12

**Authors:** Avia D. Wilkerson, Corey K. Gentle, Camila Ortega, Zahraa Al-Hilli

**Affiliations:** 1Department of General Surgery, Digestive Disease and Surgery Institute, Cleveland Clinic, Cleveland, OH 44195, USA; wilkera@ccf.org (A.D.W.);; 2Breast Center, Integrated Surgical Institute, Cleveland Clinic, Cleveland, OH 44195, USA

**Keywords:** breast cancer, disparities, prevention, diagnosis, treatment, outcomes

## Abstract

Breast cancer survival has increased significantly over the last few decades due to more effective strategies for prevention and risk modification, advancements in imaging detection, screening, and multimodal treatment algorithms. However, many have observed disparities in benefits derived from such improvements across populations and demographic groups. This review summarizes published works that contextualize modern disparities in breast cancer prevention, diagnosis, and treatment and presents potential strategies for reducing disparities. We conducted searches for studies that directly investigated and/or reported disparities in breast cancer prevention, detection, or treatment. Demographic factors, social determinants of health, and inequitable healthcare delivery may impede the ability of individuals and communities to employ risk-mitigating behaviors and prevention strategies. The disparate access to quality screening and timely diagnosis experienced by various groups poses significant hurdles to optimal care and survival. Finally, barriers to access and inequitable healthcare delivery patterns reinforce inequitable application of standards of care. Cumulatively, these disparities underlie notable differences in the incidence, severity, and survival of breast cancers. Efforts toward mitigation will require collaborative approaches and partnerships between communities, governments, and healthcare organizations, which must be considered equal stakeholders in the fight for equity in breast cancer care and outcomes.

## 1. Disparities in Breast Cancer Prevention

Breast cancer prevention entails mitigation of modifiable risk factors and identification of non-modifiable risk factors, which may warrant enhanced screening to promote early detection of cancers or high-risk lesions. Unfortunately, factors such as social determinants of health and inequitable healthcare delivery may impede the ability of individuals and communities to participate in risk-mitigating behaviors and prevention strategies, contributing to disparities in the incidence, severity, and survival of breast cancers. Modifiable risk factors for breast cancer include low physical activity, obesity, exposure to hormone replacement, and alcohol use. Non-modifiable risk factors include genetics, menstrual and birthing history, breast density, and benign breast disease. 

### 1.1. Modifiable Risk Factors

#### 1.1.1. Physical Activity

It is estimated that 25% of all cancers worldwide are related to sedentary lifestyle and obesity [1]. For breast cancer, a meta-analysis showed a 12–21% reduction in breast cancer risk among the most physically active compared with the least physically active women [2]. Breast cancer risk reduction with increased physical activity has been noted in both premenopausal and postmenopausal breast cancers [1,3,4]. Notably, the relationship between physical activity and decreased breast cancer risk appears to be independent of adiposity, with an obvious inverse relationship between physical activity and obesity [2]. Moderate to vigorous activity decreases breast cancer risk by altering cellular hormonal and stress signaling, decreasing inflammation, and enhancing immune function [5,6]. 

Though physical activity is one of the most effective tools for breast cancer risk reduction, access to and benefit from physical activity may not be equal for all individuals or populations. For example, some studies have shown that the benefit of activity-induced risk reduction is predominately in triple negative breast cancer prevention, followed by luminal A subtype breast cancers as compared to luminal B, luminal B-like, or HER2-enriched cancers [7]. Others have observed more pronounced risk reduction in post-menopausal women [8]. Moreover, others have demonstrated racial, ethnic, and socioeconomic status (SES)-related disparities in physical activity [9,10]. Leisure time physical activity has been shown to be lower among low-income populations as well as many racial/ethnic minorities [11]. Factors underlying this disparity are complex and likely include disproportionately lower access to quality environments to engage in physical activity, transportation, fitness center membership, parks, and safe streetscapes [12,13,14]. The CDC recommends six evidence-based strategies to increase physical activity, which include: (1) community-wide campaigns; (2) signs encouraging stair use; (3) health-related behavioral change programs that can be adapted to individuals; (4) school physical education programs; (5) interventions to increase social support in community settings; and (6) optimizing access to locations for physical activity combined with informational outreach [15].

#### 1.1.2. Obesity

While physical activity reduces activity of pro-oncogenic inflammatory pathways, obesity promotes a hypoxic and inflammatory cellular environment through the activity of leptin, vascular endothelial growth factors, and the infiltration of pro-inflammatory immune cells [16,17,18,19,20,21]. Further, in post-menopausal settings, estrogen biosynthesis is carried out predominately by aromatase in adipose tissue, while increased production of numerous pro-inflammatory adipokines in obese adipose tissue stimulates increased activity of aromatase, thereby increasing circulating estrogen [22,23]. Lastly, obesity is a strong contributor to metabolic syndrome, which promotes insulin resistance and hyperinsulinemia, increasing breast cancer risk and mitogenicity in breast tissue [24,25,26]. 

It thus comes as no surprise that obesity is associated with increased breast cancer incidence and worse breast cancer survival [27]. According to the CDC, post-menopausal breast cancer is the most common obesity-related cancer among women in the US [28]. Obesity is associated with a relative risk of 1.2–1.4 for post-menopausal breast cancer, with greater risk as BMI increases [29,30,31]. Further, the relationship between increased BMI, adiposity, and breast cancer risk in premenopausal women, particularly in adults who are 18–24 years of age, is poorly understood and appears to have stronger associations with hormone-positive cancers [29].

Beyond age and menopausal status, obesity also disproportionally affects women with lower education, older adults, and minority racial groups. These factors often coalesce in specific geographic areas, such as states in the southern and midwestern US where obesity rates are reported as high as 40% [32]. Obesity rates among college-educated adults were 26.3% compared to 37.8% in adults without a high school degree or equivalent. Obesity was also less common among young adults than middle-aged adults [32]. In a study finding that excess weight caused nearly 500,000 deaths in 2016, relative excess weight-related mortality was found to be higher for non-Hispanic Black adults compared to non-Hispanic White and other ethnicities and was also higher among women than men [33]. A number of studies have discussed the disproportionate burden of obesity in Black women, which carries particular significance given that Black women have the highest five-year breast cancer mortality rate of any other group, followed by Native American/Alaskan Native women [34]. It is also important to note that differential effects of obesity and increased adiposity may exist according to race and ethnicity. For example, obesity as measured by BMI seems to have a greater adverse effect in Asian women while central obesity appears to have a stronger impact on breast cancer risk in African American women [35]. As such, BMI itself may be an incomplete proxy for obesity-related breast cancer risk.

Weight reduction in obese individuals has been associated with decreased chronic inflammation and either partial or complete reversal of metabolic disturbances [36]. Weight reduction involves a combination of physical activity, diet optimization, and possibly weight-modifying medications or bariatric surgery. However, geospatial factors may limit access to whole, high-quality foods and nutrition education [37] and may disproportionately affect areas with higher concentrations of lower income individuals or ethnic minorities [38]. As such, weight management interventions must be multifaceted and include strategies targeting individuals and communities as well as government and public health sectors. Decreased lifetime risk of obesity-related cancers, including breast cancer, has been demonstrated among morbidly obese patients who underwent bariatric surgery compared to those who did not [39]. However, others have shown that access to bariatric surgery may be limited in racial minority groups, including Hispanic patients and African American men as well as those with lower incomes [40,41,42,43].

#### 1.1.3. Hormone Exposure

The relationship between estrogen and progesterone exposure and breast cancer risk is complex and increasingly debated. Modifiable risk includes exogenous hormone exposure, particularly contraceptive use in premenopausal women and hormone replacement therapy in postmenopausal women. A recent study of 1.5 million women aged 15–34 investigated breast cancer risk associated with hormonal contraceptive use. There was no increased breast cancer risk with combined hormonal contraception (IRR 1.03), though there was an increase in risk observed with progestogen-only therapies (IRR 1.32). Increased breast cancer risk was highest in the first five years and absolved ten years after discontinuation [44]. Others have found that even among those who discontinued use, risk remained higher for those who were on therapy for five or more years [45]. The Women’s Health Initiative (WHI) Hormone Trials, which included postmenopausal women aged 50–79, showed that after five years of combined estrogen and progestin therapy there was a statistically significant 26% increase in incidence of breast cancer, while the estrogen-only arm showed a 22% lower incidence of breast cancer after a mean of 10.7 years of follow up [46,47]. Regarding endogenous hormones, investigators observed evidence of increased exposure among premenopausal African American women, who were noted to have higher levels of estradiol across the menstrual cycle, even when matched for BMI, follicle dominance, and phase and cycle length [48], with 17% higher levels of circulating estradiol compared to Caucasian women [49]. The significance of these findings lies within a context of increased risk for premenopausal breast cancer in this population.

#### 1.1.4. Alcohol Consumption 

Mechanisms linking ethanol consumption and breast neoplasia remain unclear [50]. Mammary epithelial cells contain alcohol dehydrogenase that metabolizes ethanol into acetaldehyde, which accumulates in breast tissue and can form DNA adducts, point mutations, crosslinks, and other chromosomal abnormalities. Reactive oxygen species formed by ethanol metabolism can also cause oxidative DNA damage, while ethanol consumption increases intracellular estrogens, ER-alpha expression, and stimulates the proliferation of ER+ cancer cells. A meta-analysis involving 153,582 women from 53 studies found an increase in relative breast cancer risk of 7.1% for each alcoholic beverage consumed on a daily basis [51]. It also found that women who consumed 2–3 alcoholic beverages per day had a 20% higher risk of breast cancer than women who did not consume alcohol. Alcohol consumption is also an important modifiable risk factor to consider in individuals with known non-modifiable risk factors such as BRCA1 and BRCA2 mutations; these individuals are more likely to possess the ADH1C*1 allele of alcohol dehydrogenase, which may further amplify breast cancer risk due to alcohol use. Finally, geospatial disparities should also be considered in raising awareness of increased breast cancer risk with alcohol use as alcohol retailers tend to exist at significantly higher densities in lower income and underserved areas [52], such that people of lower SES may suffer a higher burden of alcohol-related consequences compared to those of higher SES [53].

### 1.2. Non-Modifiable Risk Factors

#### 1.2.1. Genetics 

According to the CDC, 5–10% of all breast cancers are hereditary [54]. Genetic predisposition to breast cancer may range from significant familial clustering of breast cancer due to underlying moderate (CHEK2, BRIPI, ATM, PALB2) or high (BRCA1, BRCA2, PTEN, TP53, CDH1, STK11) penetrance genetic mutations to multiple low penetrant alleles that contribute to risk in a polygenic fashion due to multiple single nucleotide polymorphisms. Importantly, in patients with family histories suggestive of hereditary breast cancer, predisposing genes are identified in only 30% of cases [55]. Identifying patients at risk for hereditary breast cancer is essential for early implementation of risk reduction strategies and enhanced screening. Unfortunately, numerous barriers to genetic counseling and testing have been described. For example, publicly insured patients are less likely to undergo genetic counseling than those with private insurance [56]. Similarly, lower rates of genetic testing have been reported among racial and ethnic minorities, including Black women, who are significantly more likely to be diagnosed with early onset breast cancers [57]. Underlying reasons for these discrepancies include gaps in patient education (surveys showed that White patients were 74% more likely to have heard of genetic testing than non-White patients [58]) as well as elevated concern among non-White patients regarding the use of their genetic information by the medical community. [59] However, there are also provider-related contributing factors, such as decreased provider referrals and provider-led discussions on genetic counseling for Black patients compared to White patients [60,61,62], which is particularly concerning given the higher risk of BRCA mutations in Black women [63,64,65,66,67]. The historically low participation of minority groups in genetic testing and genomic studies has biased current knowledge of hereditary breast cancer risk towards those of European ancestry. Thus, African, Asian, Latin American, and Native American groups are significantly underrepresented in genetic breast cancer databases, limiting accurate data on the prevalence of hereditary breast cancers in these groups [68]. Recent studies highlighted increased willingness and eagerness to undergo genetic counseling and equivalent uptake in genetic testing among Black women once counseled on implications for cancer prevention, and referrals were placed by their providers [69]. Better provider-led discussion with patients of all backgrounds regarding the significance and benefits of genetic testing and participation in genomic studies is needed to diversify current knowledge.

#### 1.2.2. Menstrual and Birthing History 

Menarche prior to age 12, first live birth at or after age 30, and nulliparity are associated with increased lifetime breast cancer risk. Prior investigations have revealed that after controlling for BMI, Black and Hispanic girls often experience menarche earlier than White girls [70] with reductions in these differences (50% in Hispanic women; 40% in Black women) after adjusting for SES [71]. A similar study found that women with unreliable insurance reported live birth ratios (rates of pregnancies resulting in live births) at a 6% lower rate than women with stable insurance. Similarly, Black women also experienced live birth ratios (rates of pregnancies resulting in live births) at a 9% lower rate than White women, a difference which persists even after adjusting for SES [72]. The CDC reports Black women are significantly more likely to experience still births, the cause of which is likely multifactorial [73,74]. Poorer birthing outcomes are also well documented among Native Americans and are worse in urban environments with greater healthcare access [75]. In sum, differences in menstrual and birthing outcomes in racial/ethnic minorities appear to be multifaceted, with causes including not only SES-related factors but also implicit bias, institutional racism, and psychosocial stress. 

#### 1.2.3. Breast Density 

Breast density is the ratio of breast and connective tissue to fatty tissue by mammographic assessment. Dense breast are associated with a 4–5 fold higher risk of breast cancer [76,77]. Breast density notably varies with age and BMI, such that 40–50% of women 40–44 have dense breasts compared to 20–30% of women ages 70–74 and 50–60% of women with healthy BMIs have dense breasts compared to 30% of obese women [77,78]. Postmenopausal women on hormone replacement therapy have denser breasts than those not on hormonal therapy [79]. Breast density as a risk factor for breast cancer varies by race and ethnicity. For example, the association of breast cancer risk due to higher breast density is strongest in young versus older Asian women [80]. After adjusting for age, BMI, and other cancer risk factors, Black women demonstrate significantly higher breast density in all categories (absolute area density, percent density, volume density, etc.), which is associated with increased difficulty of mammographic interpretation [81]. Because dense breasts can obscure mammographic findings of neoplasia or high-risk lesions, identification of and subsequent management of breast imaging in women with dense breasts is essential. Unfortunately, there has been documented variation in assessment and interpretation of breast density by radiologists [82] and a decreased likelihood for minority women with dense breasts to be ordered supplemental imaging compared to White women [83]. Efforts to increase streamlining identification and further imaging for women with dense breast tissue are needed to mediate such disparities. Current efforts include testing of deep learning algorithms to evaluate breast density, which have demonstrated accuracy consistent with experienced breast radiologists [84]. 

#### 1.2.4. High-Risk Benign Breast Lesions 

Atypical hyperplasia (ductal or lobular) is associated with a relative risk of breast cancer of 4.24, significantly higher than non-proliferative diseases (RR 1.27) and proliferative disease without atypia (RR 1.88) [85]. Perhaps the most relevant source of disparity in the mitigation of risk in benign breast disease is proper and timely identification and management, which may include surgical excision or even chemoprevention. Ethnic minorities and women with lower educational attainment were found to have suboptimal follow up of abnormal screening mammograms, the causes of which ranged from physician–patient miscommunication, lack of provider/health system coordination, and absence of adequate retrieval systems [86]. Access to routine screening is also critical for identifying high-risk breast lesions, which have been shown to differ by SES, education level, insurance status, access to healthcare, race, ethnicity, and gender identity [86,87,88]. In sum, managing risk portended by benign breast disease is dependent on access to screening, diagnostic imaging, and further follow-up for management. 

### 1.3. Risk Assessment 

Ultimately, assessment of patients’ cumulative BC risk via formal risk assessment provides the best opportunity for the identification of women who would benefit from enhanced or earlier screening and genetic testing to optimize prevention and early detection. A number of risk assessment tools have been developed and used over the years in the form of statistical models, such as the Gail and Tyrer-Cuzick models. However, research has identified that these models do not equitably predict risk across all populations. The Breast Cancer Risk Assessment Tool, also known as the Gail model, calculates a woman’s five-year and lifetime risk of developing breast cancer using seven variables: age, age at menarche, age at first live birth or nulliparous history, family history of breast cancer in a primary relative, history of breast biopsies, biopsies positive for atypical hyperplasia, and race/ethnicity. Some recognized pitfalls of the Gail model are the inclusion of first-degree relatives only, which may result in underestimation of risk, lack of consideration of age of onset in one’s family history, as well as personal history of breast cancer, ductal or lobular carcinoma in situ, or known genetic mutations such as BRCA1 or BRCA2. Furthermore, the model was developed from data in which racial and ethnic minorities were largely underrepresented. Such a discrepancy is critical given that the model calculates risk based on variables with breast cancer associations that may differ by race [89]. Furthermore, the model best predicts risk for ER+ cancers, the rates of which are significantly lower in younger and minority populations [89,90]. The Gail model has been shown to significantly underestimate breast cancer risk in Black women, even when revised to include data from the Black Women’s Health Study (BWHS) [91]. Conversely, both the Caucasian American and Asian American Gail models were found to give an almost two-fold overestimation of breast cancer risk in Asian women [92]. An adjusted model has also been developed for Hispanic women, with inclusion of data from the San Francisco Bay Area Breast Cancer Study, which appears to overestimate risk for Hispanic women born outside the United States versus US natives [93]. The Tyrer-Cuzick model is another frequently used risk model that integrates variables such as estrogen exposure, benign breast disease, known pathogenic genetic variance (of varying penetrance), and more extensive family history. Validation studies suggest that the Tyrer-Cuzick model offers more accurate prediction for most racial/ethnic groups but still overestimated risk for Hispanic women [94]. In 2018, the American College of Radiology updated its breast cancer screening recommendation guidelines, urging the importance of risk assessment by age 30 to allow for optimal surveillance and risk management, as they recognize that disparities in cancer risk assessment and delays in cancer identification may particularly impact outcomes for certain populations at increased risk for earlier cancers and TNBC, such as Ashkenazi Jewish women and Black women [95]. Given variability in the predictive accuracy of various assessment tools, a more recently developed model utilizing data from the BWHS has demonstrated superior performance in Black women, particularly those younger than age 40 [96]. Polygenic risk scoring (PRS), using cumulative assessment of risk based on a multitude of “common” genetic variants and SNPs, also offers opportunity for improved risk stratification for personalization of risk mitigation and screening strategies. However, PRS performance is also suboptimal in non-European populations, particularly those of African ancestry [97]. There are current efforts to develop models for polygenic risk scoring that offer equitable risk prediction [98]. 

## 2. Disparities in Breast Cancer Diagnosis

While prevention efforts work to reduce cancer risk, timely diagnosis of breast cancers provides the best opportunity for optimal outcomes and survival. Disparities in access to quality screening and interpretation of results experienced by various groups represent crucial hurdles in the establishment of equitable breast cancer care and outcomes. The COVID-19 pandemic highlighted these issues on a grand scale, yet also shed light on ways in which such disparities can be mitigated, including the importance of partnerships between community and government-backed organizations as well as use of community navigators to bridge gaps in communication and education. 

### 2.1. Breast Cancer Screening

#### 2.1.1. Disparities by Race, Insurance Status, Income, and Education

Multiple studies across the United States have demonstrated racial and socioeconomic disparities in utilization of breast cancer screening [99,100,101,102,103,104]. A systematic review and meta-analysis of 39 studies exploring utilization of screening mammography by racial groups found that among 5.8 million patients, Black and Hispanic populations had lower odds of utilizing screening mammography compared to White women (OR 0.81; CI 0.72–0.91; OR 0.83; CI 0.74–0.93, respectively), with no difference in utilization between Asians/Pacific Islanders and White women [105]. A separate national study focusing specifically on Asian-American women found that they were less likely to have had a screening mammogram in the preceding year than non-Hispanic White women (OR 0.68; CI 0.46–0.99; *p* = 0.047), with no difference in biopsy rate or breast cancer diagnosis after biopsy [102]. However, there is also evidence for misclassification [106] and over-generalization [107] of racial minorities in health data and cancer registries, which likely contributes to missed disparities. 

Another consideration regarding racial disparities in screening mammography exists at the level of screening recommendations. Given that breast cancer incidence rates are higher among Black women under age 45 compared to White women [108] and that Black, American Indian/Alaska Native and Hispanic women under age 50 are diagnosed at later stages [109], the United States Preventative Services Task Force (USPSTF) recommendation to initiate breast cancer screening at age 50 may disproportionately and negatively impact women of racial and ethnic minority groups [101], which could lead to diagnoses at later stages. Fortunately, the USPSTF released a draft recommendation this year to initiate biennial screening at age 40, and further work will be needed to assess the benefit of this new recommendation. In addition to earlier age at diagnosis among Black women, increased racial residential segregation was associated with advanced stage at diagnosis (stage III/IV) for Black women, suggesting that structural racism may contribute to decreased access to appropriate breast cancer screening within segregated areas [110]. Even among breast cancer survivors, one systematic review found 15 individual studies that reported statistically significant differences in breast cancer screening by race and ethnicity, where racial minorities (mostly Hispanic and Black women) were less likely to receive timely surveillance mammograms compared to White women [111]. 

In addition to racial factors, a large study of women in the National Health Interview Survey (NHIS) found that factors associated with being up-to-date with USPSTF breast cancer screening recommendations included education (80.4% of women with a college degree were up-to-date compared to 63% with less than a high school diploma; *p* < 0.001), federal poverty threshold (79.5% of women at >400% of the federal poverty threshold compared to 58.6% of women at <138% of the federal poverty threshold; *p* < 0.001), and insurance status (77.2% with private insurance compared to 67.2% with public insurance and 39.5% who were uninsured; *p* < 0.001) [103]. Compared to women who live in urban areas, those who live in rural areas tend to have lower educational attainment, higher rates of poverty, and lower rates of insurance coverage [112]. The further disadvantage for women who live in rural areas is that distance to the nearest screening mammography facility is greater, with travel times 4–8 fold longer to obtain breast imaging than for urban women [113]. Longer travel times have been associated with lower mammography frequency, irrespective of income and education [114]. 

#### 2.1.2. Access to Advanced Screening Technology

In addition to disparities in screening mammography rates among racial and ethnic minority groups, multiple researchers have identified disparities in access to high-quality screening services. One study of breast cancer imaging facilities in Chicago found that White women were more likely than Black or Hispanic women to receive mammograms at academic facilities, facilities that relied exclusively on breast imaging specialists to read mammograms (as opposed to general radiologists) and facilities where digital mammography was available [115]. Similar patterns were noted in women with private health insurance compared to those with other insurance types. This is notable as another study of 149 women diagnosed with breast cancer in the same city found that patients of minority racial groups with lower income and education and without private health insurance were more likely to have a missed lesion in the same quadrant as their breast cancer on blinded re-review of prior mammograms [116]. Lower use of breast imaging specialists and digital mammography could be associated with this potential missed detection of breast cancer. 

Another study of a national cohort of Medicare patients during the transition from screen-film mammography to digital mammography and digital breast tomosynthesis (DBT) found that the odds ratio for Black women receiving digital mammography compared to White women was 0.8 (CI 0.7–0.9; *p* < 0.001), and use of DBT was also less likely for both Black women and other racial/ethnic minority groups compared to White women (OR 0.84; CI 0.81–0.87; *p* < 0.001) [117]. Even in facilities that offered both DBT and regular digital mammographic screening, actual use of DBT screening among women of racial minority groups and those with lower education and income was lower than in White women and those with higher education and income [118]. Authors cited higher out-of-pocket costs or copays for DBT as potential barriers to their use among lower income patients. As DBT has been shown to reduce unnecessary recalls and increase detection of cancer, addressing barriers to DBT access and use among patients from minority racial groups and with lower education and income is critical.

#### 2.1.3. Disparities in Other Minority Groups

While many studies explore disparities in breast cancer screening by race, economic status, and educational factors, other minority and underserved groups are also affected. 

##### Immigration Status

Immigration status has been shown to impact stage at breast cancer diagnosis. In a meta-analysis of 11 studies performed in a variety of developed countries, foreign-born women were 12% less likely to be diagnosed with breast cancer at a localized stage than native-born women (OR 0.88; CI 0.82–0.95), though disparities were smaller for immigrants born in developed countries [119]. Authors postulate this disparity could be due to biological differences in cancer incidence and aggressiveness as well as differences in health care access and use. Indeed, immigrants may face additional barriers to healthcare utilization including language, cultural factors, and difficulties with navigating a foreign healthcare system. Some even argue that immigration status should be considered a social determinant of health [120]. 

##### Women with Disabilities

Women who self-reported functional impairments including movement difficulties or activity limitations have been found to undergo screening mammography at lower rates compared to women without disabilities, with disparities growing over time [121]. Additionally, one study of National Health Interview Survey (NHIS) respondents from 2010, 2013, 2015, and 2018 found that women who self-reported movement difficulties or complex activity limitations reported a lower rate of recent screening mammography and that the odds of developing breast cancer in this population compared to women without reported disability was 1.21 (CI 1.01–1.46; *p* = 0.04) [122]. 

##### Sexual and Gender Identity

Minority groups based on sexual and gender identity are also at risk for disparities in breast cancer diagnosis. One study found that compared to cisgender individuals, persons who transitioned from female to male were significantly less likely to have undergone mammographic screening (OR 0.32) [123]. A cross-sectional study of online survey participants identifying as either Black or White with self-reported abnormal breast cancer screening results or breast cancer diagnosis were asked to identify their sexual orientation as either non-heterosexual (lesbian, gay, queer, bisexual) or heterosexual and completed the nine-item-validated Healthcare System Distrust (HCSD) Scale [124]. They found that Black non-heterosexual women had the highest scores on the HCSD scale while White heterosexual women had the lowest, and they found that 72% of the disparity in HCSD between these two groups was explained by sexual minority status, while only 23% of the disparity was due to racial identity. Healthcare system distrust has been linked to poor breast cancer outcomes, including lower receipt of breast cancer screening [125]. Thus, interventions designed to increase trust in providers and healthcare systems need to target not only racism but also homophobia.

### 2.2. Breast Cancer Diagnosis after Screening

Once a suspicious lesion has been identified on a mammogram, timely workup and cancer staging is necessary to proceed with appropriate treatment. Racial minorities have been shown to experience greater delays in diagnostic workup and initiation of treatment compared to White women [126,127]. One recent study of a sample of Medicare beneficiaries found that Black women had a higher risk of delays between initial mammogram to treatment (HR 1.26, CI 1.047–1.534, *p* = 0.015) than White women, where additional exploration by age and comorbidities did not increase the risk of diagnostic or treatment delays [126].

Regarding tumor hormonal testing, a study using SEER data from 2010 to 2016 assessed the factors associated with lack of ER, PR, or HER2 testing in the diagnosis of women with breast cancer. They found that while lack of molecular testing was rare and decreased over time, racial and ethnic, age-related, poverty-level, and stage-level disparities were found. Specifically, non-Hispanic Black women had higher odds of lack of testing for HER2 status as compared to non-Hispanic White women (OR 1.15, 95% CI 1.07–1.23) after adjusting for all factors including county poverty level [128].

### 2.3. Mitigating Disparities in Breast Cancer Diagnosis

The next crucial step after identifying healthcare disparities is working to understand potential causes and trial interventions to mitigate them. This can be particularly challenging, not only because causes are interdependent and multi-factorial but also because healthcare systems will need to partner with stakeholders in the community and government agencies to effectively target breast cancer diagnosis efforts in underserved areas [100]. A group in Boston instituted an inpatient breast cancer screening program for patients with Medicaid who were overdue for a screening mammogram [129]. This involved identifying patients on General Internal Medicine services and emailing the inpatient physician suggesting they order an inpatient screening mammogram. Of 48 patients who met initial screening criteria, 39 inpatient physicians responded with 21 patients deemed eligible, 17 of which successfully completed screening mammography during inpatient admission. Of those screened, 35% never had an outpatient screening mammogram and were, on average, four years overdue for their breast cancer screening.

Due to widespread shut-downs and fear of exposure, the COVID-19 pandemic al-so contributed to a decrease in breast cancer screening rates among several minority and disadvantaged groups [130,131] and led to the decreased frequency in diagnosis of screen-detected and early-stage breast cancers compared to later-stage cancers [132]. However, when an immediate-read screening mammography program was instituted to decrease repeat office visits during the pandemic and allow for same-day diagnostic mammography, pre-pandemic racial disparities in receipt of same-day diagnostic im-aging after abnormal screening result decreased [133]. Years prior, the same group ini-tiated a same-day biopsy program after diagnostic imaging and similarly eliminated racial and insurance related-disparities in time to biopsy [134].

To mitigate rural–urban disparities in breast cancer screening attendance, another group targeted eligible women who were not up to date with breast cancer screening and living in rural counties in Indiana and Ohio. The interventions included an interactive tailored DVD mailed to the patient that explained the benefits of screening and answered specific questions as well as a patient navigator call one week after receipt of the DVD. In a randomized controlled trial involving these two interventions, the group showed that patients who received both interventions were five times more likely to become up to date with screening compared to those given the usual form of care [112].

## 3. Disparities in Breast Cancer Treatment

Though the treatment of breast cancers has become increasingly algorithmized over the past few decades compared to other solid cancers, there remain clear disparities in the application of standards of care across patient groups and populations. There are multiple barriers in health care access preventing women from obtaining optimal breast cancer treatment. It has been well established in the literature that women living in U.S. rural areas have disadvantages in terms of access to adjuvant treatment and surgical interventions. Low income, lack of insurance, and overall socioeconomic deprivation are strongly associated with delays in care that translate into poor and limited treatment adherence, ultimately affecting breast cancer survival outcomes. Racial and ethnic disparities remain a concerning factor affecting breast cancer treatment as well, which are also compounded by the fact that many minority groups are also more likely to experience socioeconomic disadvantages that translate to suboptimal breast cancer management. 

### 3.1. Hormonal Therapy 

There are well-documented differences in access to and delivery of medical and surgical treatment for breast cancer associated with patient demographics. For example, access to hormonal therapy and further adherence to hormonal regimens may be limited insurance and socioeconomic status. Nekhlyudov et al. studied the patterns of hormonal therapy use over five years in more than two thousand women diagnosed with early-stage hormone-positive receptor breast cancers under a New England health plan and found that less than one third of the patients started hormonal therapy within 12 months of diagnosis [135]. Furthermore, they reported that less than 80% of women on endocrine therapy sustained treatment without gaps (longer than 60 days) during their first year of follow-up. Over time, the persistence of treatment decreased to less than 30% of the population remaining in treatment without gaps by year five on follow-up. After adjusting for demographic and socioeconomic variables, they found a significant relationship between insurance status, lower income, and treatment adherence and found that women with longer gaps in medication were less likely to resume treatment over time. Additionally, Sood et al. [2] described similar results in a cohort of low-income women who had a pronounced lack of hormonal therapy initiation within a year of diagnosis (60%) [136]. Furthermore, of those who did initiate hormonal therapy, only 40% completed the first year of treatment and only 20% remained adherent for 5 years. They found that delays in treatment initiation were associated with decreased rates of treatment compliance, suggesting that tardiness in therapy initiation was perceived as a treatment that lacked importance and therefore negated the need to remain compliant. Others have also found differences in compliance by race/ethnicity driven by differences in severity of associated symptoms. Hu et al. found that Black women reported greater symptom burden related to adjuvant endocrine therapy, which translated to decreased adherence [137]. Given that endocrine therapy is one of the greatest contributors to improved survival in hormone-positive breast cancers, the authors concluded that better symptom management may improve adherence to therapy and diminish outcome disparities by race.

### 3.2. Adjuvant Chemotherapy and Radiation

Radiotherapy and chemotherapy are established adjuvant strategies that improve overall survival in breast cancer patients. Despite well-established benefits, barriers to accessing equitable care and treatment constitute significant contributors to mortality. Bickell et al. studied a group of 677 woman to assess receipt of adjuvant therapy and found that Black women who underwent BCT were less likely to undergo radiation [138]. Furthermore, Black women with receptor-negative cancers were less likely to receive chemotherapy than those who had hormone receptor-positive cancers. They concluded that women from minority racial groups had less probability overall of obtaining beneficial adjuvant therapy after adjusting for statistical variants. Dragun et al. also reported underuse of radiation in patients who underwent BCT and found a significant negative association in patients residing in rural areas, Medicare-insured patients, and women greater than 70 years old [139]. Additionally, the rate of receiving radiotherapy was higher in White patients when compared to Black patients. These reports demonstrate that the underuse of adjuvant breast cancer therapies in minorities, older patients, uninsured and/or rural populations may pose significant impediments in access to comprehensive breast cancer treatment and equitable cancer outcomes.

### 3.3. Targeted Therapies 

With increased investigation and insight into the pathophysiology of various breast cancer subtypes, several therapies have emerged as valuable tools in targeting unique molecular characteristics of cancers, which promote their proliferation and survival. Moreover, certain targeted therapies may improve the efficacy of more traditional breast cancer therapies such as chemotherapy and endocrine treatments [140]. Examples include monoclonal antibodies and antibody–drug conjugates in HER2+ breast cancer, CDK4/6 inhibitors in hormone-positive breast cancers, PARP inhibitors for women with BRCA mutation-positive breast cancers, anti-PD1 humanized antibodies, and antibody–drug conjugates in TNBC, to name a few [141]. Despite enthusiasm surrounding expanding opportunities for treatment offered by targeted therapies, there are also significant considerations that must be acknowledged. For example, eligibility and/or applicability of certain targeted therapies are limited by tumor genetics. However, minority populations such as Black and Hispanic women were significantly underrepresented in precision oncology trials, while White women have been overrepresented [142]. Inequality of representation in such studies may contribute to inequitable outcomes by impeding improved understanding of differences in tumor/cancer genomics, which may contribute to differences in outcomes as well as differences in genomics, which may differentially impact efficacy of targeted therapies across populations. For example, Goel et al. found that among patients who underwent Next Generation Sequencing (NGS) for metastatic breast cancer, genes with targetable variations such as PIK3CA were identified less frequently in Black Women than in White women [142]. Nevertheless, others have found that even when actionable molecular targets such as the HER2 receptor are identified, there are significant differences in administration of targeted therapy by race even after controlling for other factors such as tumor characteristics, socioeconomic status, and comorbidities [143]. It should also be noted that survival benefit from targeted therapies was demonstrated to be significant for more advanced cancers, which are already associated with higher cost and financial burden [144,145]. As such, cost associated with administration of targeted therapies in the scope of precision medicine may systematically impede access to and amplify financial toxicity for patients with low SES, low income, or no insurance. Further, cost-related barriers in access to targeted treatments may also differentially impact minority populations that are more likely to present with later stage disease. 

### 3.4. Surgical Treatment

Disparities in healthcare also impact the surgical management of breast cancer. Dankwa-Mullan et al. studied a large population of insured women with non-metastatic breast cancer to assess for clinical and non-clinical factors determining the use of breast conservation therapy (BCT) [146]. They reviewed 53,060 patients who had surgery within 6 months of diagnosis and found that 68% underwent breast conservation surgery and 32% underwent mastectomy. The study noted that patients with higher household incomes had an increased likelihood of BCT; this was also true for patients residing in urban areas with greater access to gynecologists, geneticists, and plastic surgeons, suggesting that a multidisciplinary approach leads to increased rates of BCT use. Similarly, Lautner et al. studied a large cohort of almost 730,000 women to characterize the relationship between patient demographics and likelihood of receiving BCT [147]. They described an increased use of BCT in women with higher incomes and education levels, as well as in women with private insurance. Their study also assessed the influence of the location, type of institution (academic vs. community), and travel distance to the treatment facility and found that the lowest rate of BCT use happened in community cancer centers and among patients who traveled distances to their facility greater than 27.8 km, the latter suggesting that access to standard radiotherapy in urban vs. rural areas is variable and directly influences the use of BCT. A multitude of studies have demonstrated delays in time to treatment, including time to surgery by race [148,149]. In fact, a National Cancer Database Study by Jackson et al. showed that 30.6% of non-Hispanic Black women had surgery more than 60 days after diagnosis compared to only 18% of White women [150].

### 3.5. Management of the Axilla

Sentinel lymph node biopsy (SLNB) is the preferred method for axillary staging in patients with clinically node-negative breast cancer and unfortunately has not been exempt from inequitable application since its development. Black et al. reported higher rates of SLNB in patients with higher education levels and incomes and significantly lower rates of use in African American women when compared to Caucasians [151]. Given that type of surgery was found to independently predict the use of SLNB, the group completed a stratified analysis to determine if disparities continued despite surgical intervention and showed that African American women were in fact less likely to receive SLNB regardless of undergoing lumpectomy or mastectomy when compared to White patients. Such differences in management have significant implications on morbidity and postoperative outcomes, specifically in regard to lymphedema rates that can be as high as 30% depending on the extent of axillary surgery. Similarly, Reeder-Hayes et al. [8] evaluated the differences in SLNB use by age and race in a cohort of early-stage breast cancer patients [152]. They reported that African American women were significantly less likely to undergo SNLB as their primary staging procedure, approximately half as likely when compared to their Caucasian counterparts. Lastly, Lautner et al. also described that a higher nodal stage at diagnosis was directly associated with a lower educational level, uninsured status, and a lower probability of undergoing BCT [147].

### 3.6. Breast Reconstruction

Since 1998, with the approval of the federal Women’s Health and Cancer Rights Act and subsequent state laws that mandated insurance coverage for breast cancer-related reconstruction, the rates of reconstruction increased [153]. However, despite these laws, access is still disparate. Shippee et al. analyzed the association between race and insurance status for reconstruction rates in a very large cohort of women under age 65 [154]. They found that reconstruction rates were higher in patients who were White, younger than 45 years old, and in those with private insurance coverage. They also reported that women undergoing treatment in urban areas were almost three times more likely to undergo reconstruction than women treated in rural hospitals. Similarly, Soni et al. discussed, in their review, that African American women were less likely overall to undergo breast reconstruction than White women after adjusting for additional demographic variables [155]. These disparities in care have meaningful implications since inequitable access to reconstruction potentially excludes the population from the established psychosocial and clinical benefits.

## 4. Disparities in Survivorship

Cancer survivorship refers to the experiences and processes by which someone progresses through diagnosis and treatment, onto life after cancer treatment, and beyond. Some have proposed phases of survivorship such as: (1) diagnosis with early-stage cancer with goals of treatment set upon intent to cure; (2) diagnosis or progression to advanced or metastatic cancer with goals of treatment set upon prolonging life; and (3) diagnosis or progression to end-stage cancer with goals of care centered on end-of-life care [156]. The experiences of survivors may be characterized by various facets ranging from physical to psychological to social issues. As breast cancer survivorship is irrevocably tied to survival, it is unsurprising that investigators have identified disparities in survivorship by race. Groups identified to have worse breast cancer survival have also been found to have worse experiences related to survivorship. For example, Black women have been found by multiple investigative groups to experience worse long-term and late effects of breast cancer treatment, such as a two-fold higher risk of lymphedema, more frequent taxane-induced peripheral neuropathy (often translating to less receipt of chemotherapy), and more frequent HER2 therapy-associated cardiotoxicity [157,158,159]. Similarly, African American women living with breast cancer in urban areas were fifty percent more likely to lose employment after diagnosis than White women, also highlighting potential differences in social stresses and financial toxicity [160]. Partners of Hispanic breast cancer survivors were also more likely to report worsened financial status after their partners’ breast cancer diagnosis than their White counterparts [161]. Hispanic women with breast cancer reported worse quality of life according to physical, mental, and social health-related issues than any other group [162]. Others have shown that, after accounting for demographics (including race) and clinical information, socioeconomic status correlates significantly with quality of life related to physical and mental health; of 703 multiethnic breast cancer survivors, those with higher socioeconomic status reported better quality of life [163]. More research is needed to expand on survivorship disparities by factors such as age at diagnosis, geography, immigration status, and, plausibly, breast cancer subtype. 

## 5. Conclusions

Advances in standardized screening and improved diagnosis and treatment of breast cancer have substantially improved survival rates over the past few decades. However, the impact of such advancements has not equivalently translated to optimized survival in all women. The first consideration is that some women at high risk of developing breast cancer are not being appropriately treated with enhanced screening and counseling, including those with identified genetic risk, early menarche, dense breast tissue, and high-risk benign breast lesions. The next consideration is modifiable risk factors for breast cancer disproportionately impacting patients according to education level, race/ethnicity, residential location, and medical insurance status. These modifiable risk factors include physical activity, obesity, exogenous hormone exposure, and alcohol use, which all contribute to a further increase in breast cancer risk. The national scope for such factors is far-reaching and complex and will likely necessitate collaborations and partnerships between primary care providers, community and governmental organizations, and high-risk breast disease specialists to help raise awareness about the importance of risk assessment and promote appropriate patient referral to high-risk breast cancer clinics and centers for genetic counseling and screening.

Timely diagnosis of breast cancers provides the best opportunity for optimal outcomes and survival. Numerous investigators have demonstrated disparate access and timely utilization of breast cancer screening, diagnostic imaging, and procurement of tissue biopsies, which affect a wide range of women in disadvantaged and minority groups. However, there are several cases in which interventions were trialed and found to mitigate or eliminate disparities in breast cancer diagnosis, offering hope and future direction to efforts in this area. These efforts include the use of community navigators to serve as community liaisons who may offer essential education on screening and follow-up as well as establishment of immediate-read screening mammography programs and same-day biopsy programs after diagnostic imaging to eliminate delays in initiation of care that disproportionately affect disadvantaged and minority populations. 

Many have investigated and reported barriers to access and discrepancies in healthcare delivery, which have translated to suboptimal application of standards of breast cancer care across populations and groups. It has been well studied that women living in rural areas face difficulties in accessing adjuvant treatment and surgical interventions. Lack of insurance and overall socioeconomic deprivation are strongly associated with delays in care that translate into poor and limited treatment adherence, ultimately affecting breast cancer survival outcomes. Racial disparities remain a concerning factor affecting breast cancer treatment as well, particularly when considering that minorities are disproportionately affected by socioeconomic disadvantages, which amplify barriers in access to optimal care. Overall, the etiologies of various disparities in breast cancer outcomes are complex and multifactorial; it is paramount to acknowledge that intersectionality of multiple factors discussed in this paper may impact the experiences and survival of breast cancer patients in a cumulative manner, particularly those from marginalized groups. 

Future directions in breast cancer disparities research must include discussions on action items regarding policy changes and efficacious interventions by which disparities may be eliminated. For example, may a greater effort to emphasize culturally appropriate cancer care be an important next step in mitigating outcome disparities by race/ethnicity? Regarding the role of language barriers in the quality of breast cancer care, Karliner et al. found that of a cohort of oncologists and oncologic surgeons, 56% reported having less patient-centered treatment discussions with patients of limited English proficiency and greater than 50% reported experiencing difficulty with conversations regarding treatment options and prognosis with such populations. These difficulties were reduced by 53% with use of professional medical interpreters, deeming this an actionable intervention providers can take to improve outcomes among immigrant patient populations [164]. More work is needed to investigate the potential role of improved cultural competency in optimizing care for non-immigrant ethnic minorities. Lastly, lessons learned from observed breast cancer disparities in the United States may also shed light on outcome differences in international contexts, particularly regarding access to cancer prevention, screening, and treatment. Even more work is needed to strategize solutions to optimize breast cancer outcomes in developing countries with greater resource unavailability.

## Data Availability

No new data were created or analyzed in this study. Data sharing is not applicable to this article.

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
