# Peer review of "Disparities in Breast Cancer Care—How Factors Related to Prevention, Diagnosis, and Treatment Drive Inequity"

_healthcare, 2024, doi:10.3390/healthcare12040462_

Round 1

Reviewer 1 Report

Comments and Suggestions for Authors

Wilkerson et al. reviewed disparities in Breast Cancer Care-How factors related to prevention, diagnosis, and treatment drive inequity. This manuscript is interesting and organized well. I have some minor concerns that need to be addressed before publication.

  1. The authors should include information about drugs that are running in clinical trials for the management of breast cancer.
  2. The authors should also discuss in detail the side effects of the currently available drug treatments for breast cancer.
  3. A future direction for the advanced management of breast cancer should be included.
  4. The molecular mechanism of breast cancer should be displayed by at least one good picture.
  5. Typo and grammatical errors should be corrected.
Comments on the Quality of English Language

Minor editing of English language required

Author Response

Response to Reviewer 1 Comments

  1. The authors should include information about drugs that are running in clinical trials for the management of breast cancer.

While this is an important consideration for practitioners and patients undergoing breast cancer treatment now, the medications under clinical trial is outside the scope of this review.

  1. The authors should also discuss in detail the side effects of the currently available drug treatments for breast cancer.

Again, while this is important for patients undergoing breast cancer treatment using available drug therapies, it is out of the scope of this review which focuses on disparities in breast cancer care.

  1. A future direction for the advanced management of breast cancer should be included.

We would need more clarification regarding this comment to adequately address it in revisions. Future directions we think are important when discussing disparate care focus more on increasing studies that explore interventions which mitigate disparities.  

  1. The molecular mechanism of breast cancer should be displayed by at least one good picture.

We feel that a figure like this would be better suited for an article which generally discusses breast cancer and its molecular underpinnings. This review article focuses on disparities in breast cancer care, and while there are differences in breast cancer prevention, incidence, diagnosis, and treatment between minority groups, a specific molecular mechanism for this is not discussed in this paper.

  1. Typo and grammatical errors should be corrected.

Thank you, these have been reviewed and corrected.

Reviewer 2 Report

Comments and Suggestions for Authors

The article titled "Disparities in Breast Cancer Care – How Factors Related to Prevention, Diagnosis, and Treatment Drive Inequity" by Avia D. Wilkerson et al. offers valuable insights. However, there are several issues that require attention:

1. The authors should provide their own justification for the study, as the relevance of the topic has already been explored in previous publications. Some of these publications can be found in PubMed, such as Gynecol Clin North Am. 2022 Mar;49(1):149-165; BMC Cancer. 2013 Oct 2;13:449; Breast Cancer Res Treat. 2010 Jul;122(2):533-43; Semin Oncol Nurs. 2015 May;31(2):170-7; Cancer Epidemiol Biomarkers Prev. 2009 Jan;18(1):121-31; J Am Coll Surg. 2023 Jun 1;236(6):1233-1239; NPJ Breast Cancer. 2015 Oct 14;1:15013, among others. Therefore, the study does not provide any innovative information.

2. The authors do not provide specific data or references to support the claims made regarding disparities in breast cancer prevention, diagnosis, and treatment. Including specific studies or statistics would enhance the credibility and reliability of the information presented.

3. The authors did not delve into a detailed discussion of the underlying causes of these disparities or explore the complex interplay between these factors.

4. Although the authors stated potential strategies for reducing disparities, they did not elaborate on what these strategies might entail. Providing more specific examples of interventions or policy changes that have shown promise in addressing disparities would be beneficial.

5. The authors can highlight the importance of ongoing research to underscore the significance of this issue and the need for evidence-based interventions.

6. The authors did not sufficiently address the concept of intersectionality, which recognizes that individuals may face multiple forms of discrimination and disadvantage simultaneously. Considering how factors such as race, socioeconomic status, and access to healthcare intersect and compound disparities in breast cancer prevention, diagnosis, and treatment would be valuable.

7. The authors predominantly focus on disparities within certain populations or demographic groups, but they do not acknowledge the global nature of breast cancer disparities. There are significant disparities in breast cancer outcomes between high-income and low-income countries, as well as within countries with varying healthcare systems.

8. The authors can address language barriers, lack of culturally appropriate information, and cultural beliefs and norms that affect healthcare-seeking behaviors.

9. The authors can include the addressing of disparities in survivorship. Factors such as access to follow-up care, rehabilitation services, and psychosocial support can significantly impact survivorship outcomes.

10. Elaborating on the role of policy changes and systemic reforms would strengthen the recommendations provided.

Author Response

Response to Reviewer 2 Comments

  1. The authors should provide their own justification for the study, as the relevance of the topic has already been explored in previous publications. Some of these publications can be found in PubMed, such as Gynecol Clin North Am. 2022 Mar;49(1):149-165; BMC Cancer. 2013 Oct 2;13:449; Breast Cancer Res Treat. 2010 Jul;122(2):533-43; Semin Oncol Nurs. 2015 May;31(2):170-7; Cancer Epidemiol Biomarkers Prev. 2009 Jan;18(1):121-31; J Am Coll Surg. 2023 Jun 1;236(6):1233-1239; NPJ Breast Cancer. 2015 Oct 14;1:15013, among others. Therefore, the study does not provide any innovative information.

Thank you for your concern regarding the justification for this study. The first study you list primarily focuses on racial disparities examining US mortality and incidence data, with brief explorations by US geography. Our article also includes an extensive review of disparities in breast cancer prevention, with a more organized approach to detection and treatment and focus on a wider number of contributing factors than just race/ethnicity. The study you list published in JACS is a brief article which only focuses on breast cancer screening, genetic counseling, fertility preservation, and breast reconstruction. The rest of your listed studies were published in or before 2015, thus our article represents an update to those with almost 10 more years of data. Thus, our review article based on its content and up-to-date review of the status of breast cancer disparities does contribute new insights to the body of literature on this topic.

  1. The authors do not provide specific data or references to support the claims made regarding disparities in breast cancer prevention, diagnosis, and treatment. Including specific studies or statistics would enhance the credibility and reliability of the information presented.

Our article includes 163 references, all of which serve to support our claims regarding disparities in breast cancer prevention, diagnosis, and treatment.

  1. The authors did not delve into a detailed discussion of the underlying causes of these disparities or explore the complex interplay between these factors.

The underlying causes of disparities are complex and include multiple different factors. We do discuss this in the body of the paper but more specifically in the conclusion. We have also expanded on this topic in the paper's conclusion after revisions.  

  1. Although the authors stated potential strategies for reducing disparities, they did not elaborate on what these strategies might entail. Providing more specific examples of interventions or policy changes that have shown promise in addressing disparities would be beneficial.

The trouble is that few studies have trialed interventions that have mitigated disparities in breast cancer care. Our Screening section does include several, one of which trialed an inpatient screening program as well as same-day diagnostic imaging and biopsy programs which were shown to mitigate or eliminate disparities in these areas. This is an area that needs more focus with future research, as much of the body of literature on this topic focuses on identifying disparities only.

  1. The authors can highlight the importance of ongoing research to underscore the significance of this issue and the need for evidence-based interventions.

We agree and have added this to the future directions section of our review article.

  1. The authors did not sufficiently address the concept of intersectionality, which recognizes that individuals may face multiple forms of discrimination and disadvantage simultaneously. Considering how factors such as race, socioeconomic status, and access to healthcare intersect and compound disparities in breast cancer prevention, diagnosis, and treatment would be valuable.

We have expanded on this topic in the conclusion section of our review article.

  1. The authors predominantly focus on disparities within certain populations or demographic groups, but they do not acknowledge the global nature of breast cancer disparities. There are significant disparities in breast cancer outcomes between high-income and low-income countries, as well as within countries with varying healthcare systems.

While there are certainly disparities in breast cancer care around the world, the focus of this article was on the status of disparate care in the United States.

  1. The authors can address language barriers, lack of culturally appropriate information, and cultural beliefs and norms that affect healthcare-seeking behaviors.

We have added this to the review article, but also already commented on it in the screening section as these factors are known to contribute to disparities in care.

  1. The authors can include the addressing of disparities in survivorship. Factors such as access to follow-up care, rehabilitation services, and psychosocial support can significantly impact survivorship outcomes.

This section has been added to our review article.

  1. Elaborating on the role of policy changes and systemic reforms would strengthen the recommendations provided.

This has been added to the conclusion section.

Reviewer 3 Report

Comments and Suggestions for Authors

Strenght.

- The paper is well written

- It is easy to follow in the logical construction in paragraphs

- It covers many epidemiological aspects of breast cancer and the relative inequalities among different social classes/etnicities, in a extensive way

- Even if briefly, it covers the main risk factors for breast cancer in a complete way

Weaknesses.

- the paper only covers epidemiological aspects, giving just a photograph of the current situation based on literature reports

- the different epidemiological aspects are just briefly covered, but this matter is too extensive to be completely covered in a paper

- Line 28: correct canscer

Author Response

Response to Reviewer 3 Comments

  1. The paper only covers epidemiological aspects, giving just a photograph of the current situation based on literature reports. The different epidemiological aspects are just briefly covered, but this matter is too extensive to be completely covered in a paper.

It is true that this is an extensive and nuanced topic, we do provide a thorough review including not only diagnosis and treatment but also prevention but recognize that no review article could completely cover this topic.

  1. Line 28: correct canscer

This mistake has been corrected.

Reviewer 4 Report

Comments and Suggestions for Authors

Dear Authors,

It is known that there are disparities in access to the whole continuum of patient care for breast cancer from prevention to therapy. This work very nicely goes through all aspects or elements of breast cancer care and takes together data that are the main factors that have an impact on access the care. In my opinion, it is a well-structured compendium of relevant data.

My critical comments:

1. This work lists studies from the United States and is therefore mostly applicable to American settings. At the same time, it can facilitate the initiation of similar investigations in other regions or countries. It might be mentioned in the conclusion.

2. row 395-401: this is not evidence of racism or homophobia. It also can be explained by other factors, as later discussed in the manuscript, e.g. lower economic or educational status. I think it is not beneficial to single out one or two factors ("healthcare systems need to target not only racism but also homophobia") that should be avoided during healthcare management. Equal treatment is not only a professional but also an ethical requirement. It must prevail regardless of all factors.

3. row 405-406: "Racial minorities have been shown to experience greater delays in diagnostic workup and initiation of treatment compared to White women." - this statement is true, but I think it should be emphasized that this is a multifactorial story. Other factors must be considered, as racial minorities have lower educational and economic status or type of insurance. In conclusion, I recommend emphasizing that the reasons are multifactorial and present not only on the provider's side but also on the patient's side, as it is present in row in rows 420-423 ("This can be particularly challenging as not only are causes interdependent and multi-factorial, but healthcare systems will need to partner with stakeholders in the community and government agencies to effectively target breast cancer diagnosis efforts in underserved areas").

Also, a good example is in rows 415-416 ("Specifically non-Hispanic Black women had higher odds of lack of testing for HER2 status as compared to non-Hispanic White women"). It is highly improbable that the pathologist would know at the microscope the race of the patient.

Author Response

Response to Reviewer 4 Comments

  1. This work lists studies from the United States and is therefore mostly applicable to American settings. At the same time, it can facilitate the initiation of similar investigations in other regions or countries. It might be mentioned in the conclusion.

This has been added to the conclusion section.

  1. row 395-401: this is not evidence of racism or homophobia. It also can be explained by other factors, as later discussed in the manuscript, e.g. lower economic or educational status. I think it is not beneficial to single out one or two factors ("healthcare systems need to target not only racism but also homophobia") that should be avoided during healthcare management. Equal treatment is not only a professional but also an ethical requirement. It must prevail regardless of all factors.

The evidence presented was not meant to demonstrate the evidence of racism or homophobia, just illustrate that these could be underscoring healthcare distrust in these populations. We agree that equal treatment is an ethical requirement, but widespread disparities highlight that these do not prevail regardless of all factors.

  1. row 405-406: "Racial minorities have been shown to experience greater delays in diagnostic workup and initiation of treatment compared to White women." - this statement is true, but I think it should be emphasized that this is a multifactorial story. Other factors must be considered, as racial minorities have lower educational and economic status or type of insurance. In conclusion, I recommend emphasizing that the reasons are multifactorial and present not only on the provider's side but also on the patient's side, as it is present in row in rows 420-423 ("This can be particularly challenging as not only are causes interdependent and multi-factorial, but healthcare systems will need to partner with stakeholders in the community and government agencies to effectively target breast cancer diagnosis efforts in underserved areas").

We have expanded on the multifactorial components of this topic in the conclusion section.

  1. Also, a good example is in rows 415-416 ("Specifically non-Hispanic Black women had higher odds of lack of testing for HER2 status as compared to non-Hispanic White women"). It is highly improbable that the pathologist would know at the microscope the race of the patient.

Our interpretation of this study result is not that pathologists know the race of the patient and do not test for HER2, but that for an unknown reason Black women are not having HER2 testing on their tumors despite this being the standard of care. The study serves to underscore another facet of this problem.

Round 2

Reviewer 2 Report

Comments and Suggestions for Authors

Accept in present form

Comments on the Quality of English Language

 Minor editing of English language required